# Cervical Screening in North Sardinia (Italy): Genotype Distribution and Prevalence of HPV among Women with ASC-US Cytology

**DOI:** 10.3390/ijerph19020693

**Published:** 2022-01-08

**Authors:** Narcisa Muresu, Giovanni Sotgiu, Silvia Marras, Davide Gentili, Illari Sechi, Andrea Cossu, Arianna Dettori, Roberto Enrico Pietri, Luisa Paoni, Maria Eugenia Ghi, Maria Paola Bagella, Adriano Marrazzu, Antonio Cossu, Antonio Genovesi, Andrea Piana, Laura Saderi

**Affiliations:** 1Department of Medical, Surgical and Experimental Sciences, University of Sassari, Padre Manzella Street, 07100 Sassari, Italy; narcisamuresu@outlook.com (N.M.); ma.silvia.87@gmail.com (S.M.); davide.gentili@gmail.com (D.G.); illarisechi@yahoo.it (I.S.); andreacossu@uniss.it (A.C.); piana@uniss.it (A.P.); 2Clinical Epidemiology and Medical Statistics Unit, Department of Medical, Surgical and Experimental Sciences, University of Sassari, Padre Manzella Street, 07100 Sassari, Italy; lsaderi@uniss.it; 3Biomedical Science Ph.D. School, Biomedical Science Department, University of Sassari, Padre Manzella Street, 07100 Sassari, Italy; ariannadettori01@gmail.com; 4ASSL Sassari, Coordinamento Consultori Familiari ASSL, Sassari 07100, Italy; robertoenrico.pietri@atssardegna.it (R.E.P.); luisa.paoni@atssardegna.it (L.P.); mariaeugenia.ghi@atssardegna.it (M.E.G.); mariapaola.bagella@atssardegna.it (M.P.B.); adriano.marrazzu@atssardegna.it (A.M.); 5Department of Medical, Surgical and Experimental Sciences, Institute of Pathology, University of Sassari, Via Matteotti, 07100 Sassari, Italy; cossu@uniss.it; 6Department Health Education, Prevention, and Health Promotion Activities, 07100 Sassari, Italy; antonio.genovesi@atssardegna.it

**Keywords:** human papillomavirus, cervical carcinoma, prevention, screening

## Abstract

The assessment of human papillomavirus (HPV) genotype dynamics could support the adoption of more tailored preventive actions against cervical cancer. The aim of the study was to describe the prevalence of HPV infection, HPV genotype distribution, and the epidemiological characteristics of women with ASC-US cytology since the introduction of HPV-DNA testing in Sardinia (Italy), (March 2016–December 2020). Specimens were tested by RT-PCR for 14 high-risk HPV genotypes. A total of 1186 patients were enrolled, with a median (IQR) age of 41 (38–48) years. Of these women, 48.1% were positive for at least one HPV genotype; 311 (26.2%) women were vaccinated with a median (IQR) age of 38 (30/47) years. The percentage of prevalence of HPV-16, -31, -66, -56, and -51 was 36.3%, 18.7%, 11.9%, 11.4% and 10.7%, respectively. The highest prevalence of infection was found in women aged <41 years, and single women. Moreover, women aged >41 years (OR: 0.51, 95% CI: 0.31–0.86; *p*-value: 0.01), having parity (OR: 0.57, 95% CI: 0.34–0.96, *p*-value: 0.04), and higher educational level (OR: 0.39, 95% CI: 0.18–0.87; *p*-value: 0.02) were associated with a lower CIN2+ risk. We did not find a significant difference in terms of prevalence of HPV-16 infection between vaccinated and non-vaccinated (18.3% vs. 17.1%; *p*-value < 0.001). Our results support the adoption of nonavalent HPV-vaccine to prevent the most prevalent infections caused by HPV-16 and -31 genotypes and underscore the need of surveillance to implement tailored vaccination programs and preventive strategies.

## 1. Introduction

Cervical cancer is the fourth most prevalent cancer in women worldwide; its annual incidence and mortality were equal to ~641,127 cases and >341,831 deaths in 2020 [1]. Despite the improved early detection and treatment, geographical differences were reported, with the highest incidence and mortality rates in Sub-Saharan Africa and Southeast Asia [1]. Persistent infection with high-risk Human Papillomavirus (Hr-HPV), mostly with HPV-16 and -18 genotypes, is the main mechanism behind the occurrence of cervical cancer [2,3]. Up to 75% of all women are exposed to HPV during their lifetime course, but the majority develops an effective immune response with a viral clearance within 2 years [4]. Adoption of preventive strategies (i.e., vaccination and cervical screening) have significantly reduced the burden of HPV-related diseases [5]: HPV-DNA testing followed by cytology increases sensitivity for severe lesions (i.e., cervical intraepithelial neoplasia CIN3 and CIN3+) and cervical carcinoma by 30%, allowing the extension of the screening interval up to 6 years, [6,7]. The screening adherence in Italy is higher than 80%, with slight regional differences. Sardinia, an Italian region of about 1.5 million inhabitants, showed a high prevalence of HPV-16 and -51 genotypes [8].

The estimation of the prevalence of HPV infections, as well as the distribution of HPV genotypes following the vaccination of young and adult cohorts, could tailor future preventive strategies.

The aim of the present study was describing the prevalence of Hr-HPV genotypes in a cohort of women attending a Sardinian (Italy) regional screening program, from March 2016 to December 2020; moreover, epidemiological factors associated with clinical progression of cervical lesions were evaluated.

## 2. Materials and Methods

### 2.1. Study Design

The study participants were recruited in Sassari, northern Sardinia (Italy). The regional screening program includes the Pap (Papanicolaou) test followed by the HPV-DNA testing in case of ASC-US (atypical squamous cell of undetermined significance), ASC-H (atypical squamous cells, cannot exclude high-grade squamous intraepithelial lesion), and ACG (atypical glandular cells) results [9]. Based on this algorithm, all women aged from 25 to 64 years with a diagnosis of ASC-US between March 2016 and December 2020 who underwent an HPV-DNA testing were enrolled.

### 2.2. Sample Collection and HPV Test

Cervical specimens were collected with cervix-brush and suspended in a 20 mL preservation solution, PreservCyt transport medium (ThinPrep Pap Test; Cytyc Corporation, Boxborough, MA, USA). A liquid-based cytology was performed for all participants: those diagnosed with ASCUS according to the TBS-2001 classification [10] underwent a single HPV-DNA test.

Nucleic acid extraction was performed using a commercially available extraction kit QIAamp DNA Mini Kit, (Qiagen, Hilden, Germany) or GeneAll RibospinTM vRD II (GeneALL, Dongnam-ro, Songpa-gu, Seoul, South Korea) [11,12].

HPV genotyping was conducted using the commercial kit Anyplex II HPV HR detection kit (Seegene Inc., Seoul, Korea) [13], a multiplex real-time polymerase chain reaction assay to detect 14 Hr-HPV (HPV-16, -18, -31, -33, -35, -39, -45, -51, -52, -56, -58, -59, -66, -68) types in a single tube.

### 2.3. Statistical Analyses

Microsoft Excel software (Microsoft Corporation, Redmond, DC, USA) was used to collect demographic, epidemiological, clinical, and virological variables. Qualitative variables are described with absolute and relative (percentage) frequencies, whereas quantitative variables are summarized with means (standard deviations, SD) or medians (interquartile ranges, IQRs), depending on their parametric distribution. Comparisons between vaccinated and non-vaccinated patients were performed with a chi-squared or Fisher’s exact test for qualitative variables. Logistic regression analysis was carried out to assess the relationship between a severe cervical lesion (CIN2+) and demographic, epidemiological, and clinical covariates.

A two-tailed p-value less than 0.05 was deemed statistically significant. All statistical analyses were performed with the statistical software STATA version 17 (StataCorp, College Station, TX, USA).

## 3. Results

A total of 1217 patients underwent HPV-DNA testing during the period March 2016 and December 2020. The median (IQR) age of the cohort was 41 (31–48) years. The majority were single (496/1186; 41.8%) and ~27% (324/1186) were married. The majority attended high and middle school (28.9% and 20.8%, respectively). More than half (231/394; 58.6%) were administered oral contraceptives and 8.9% (105/1186) had undergone a gynecological intervention in the recent past (Table 1).

Three hundred and eleven (26.2%) women were vaccinated with a first HPV vaccine dose when their median (IQR) age was 38 (30–47) years and >66% and 27.7% were vaccinated with the tetravalent and nonavalent vaccine, respectively.

HPV-DNA test was positive for at least one Hr-HPV genotype in almost half of the cases (571/1186; 48.1%), with 15% (178/1186) showing infections caused by multiple genotypes.

The most prevalent genotypes were HPV-16 (207/571; 36.3%), HPV-31 (107/571; 18.7%), HPV-66 (68/571; 11.9%), HPV-56 (67/571; 11.7%), HPV-51 (61/571; 10.7%), and HPV-39 (59/571; 10.3%) (Table 2). No statistically significant differences were observed in terms of positivity rate by age group, except for HPV-16 (*p*-value: 0.01) (Figure 1) (Appendix A). However, a higher prevalence of infection was reported in women aged <41 years (59.6% vs. 38.7%; *p*-value < 0.001).

The most frequent genotypes in multiple infections were HPV-31 and -66.

A higher prevalence of single infections was found in individuals with a negative cytology (~68%), and in those classified as CIN1 and CIN3 (57% and 71%, respectively). Infections caused by more than one genotype were mainly found in CIN2 cases (>53%).

A total of 617 women were followed-up and underwent a second pap-test after a median (IQR) period of 7 (8–9) months. Overall, most women registered a regression of lesion with a negative result (412/614; 67.1%), whereas 24.2% (149/614) and 3.6% (20/614) were classified as LSIL and HSIL, respectively.

At baseline, a lower prevalence of infection was found in married women (31.5% vs. 59.3%; *p*-value < 0.001), whereas a higher prevalence was observed in women with a higher educational level (56.8% for degree; *p*-value: 0.004). A statistically significant higher prevalence was described in women exposed to oral contraceptives (52.4% vs. 35%; *p*-value: 0.001).

Vaccinated women showed a higher positivity rate (80.4% vs. 36.7; *p*-value < 0.001). A stratified analysis on HPV-16, the most prevalent and preventable genotype included in the vaccine, did not show a difference between the two groups (18.3% vs. 17.1%; *p*-value: 0.64) (Appendix A).

Overall, a low risk of CIN2+ was found in women aged >41 years (OR: 0.551, 95%CI: 0.31–0.86; *p*-value: 0.01). Moreover, a higher educational level seems to play a protective role (OR: 0.39, 95% CI: 0.18–0.87; *p*-value: 0.02), as well as parity (OR: 0.57, 95% CI: 0.34–0.96, *p*-value: 0.04) (Table 3).

## 4. Discussion

The present study reported the results of HPV genotype distribution in women recruited into the cervical cancer screening program in Northern Sardinia, Italy, since the introduction of HPV-DNA testing in March 2016.

The overall HPV prevalence in women with ASC-US cytology was ~48%, in accordance with other national and international surveys [14,15], and slightly lower than Kjær and Colleagues who reported on a prevalence >70% in individuals with ASCUS and LSIL cytology [16]. The different prevalence estimates could be associated with the recruited participants, algorithm of screening, and geographical area. HPV prevalence, mostly for ASCUS lesions, can depend on age: a higher infection rate was found in women < 41 years aged (59.6% vs. 38.7%), mainly due to changes in sexual habits and the spontaneous clearance of previous infections [17].

Similar to other sexually transmitted diseases, the major HPV risk factors are related to sexual behaviors: age of the first sexual intercourse, number of sexual partners and habits [18]. We found an increased infection rate in single women and in those with a higher educational level. Furthermore, a higher prevalence of infection was described in women exposed to oral contraceptives. Although their role was not adequately explained, Gierish and colleagues showed an association between risk of cancer and duration of contraceptive therapy, mostly in HPV-positive women [19]; cervical ectopia can occur and, consequently, exposure of squamo-columnar tissue to viral and bacterial infections can favor cellular proliferation following estrogen and progestin stimulation [20].

HPV-16 was confirmed as the most prevalent genotype, as previously reported for other HPV-related diseases [21]. It was confirmed the high circulation of some preventable genotypes (i.e., HPV-16, and -31) in this Italian region, supporting the recommendation of the nonavalent HPV-vaccine to protect against these genotypes [8,22]. The prevalence of infection increases with the severity of the disease [23]. HPV-16 prevalence among women with low-grade cervical lesions ranged between 7.5% [24] and 36.7% [25] in Italy. Bruni and colleagues found that HPV-16 was the most common genotype, followed by HPV-52, -51, -31, -53 and -66 [23].

In a Swedish study, HPV types 16, 18, 31, 33, 45, and 52 were found in 689 of 808 screened invasive cervical cancers (85.3%). The addition of HPV types 35, 39, 51, 56, 58, 59, 66, and 68 (also included in currently used HPV tests) increased prevalence by only 12 of 808 cases (1.5%, for all these eight types together) [26]. HPV screening tests might perform better if restricted to the seven HPV types in the nonavalent vaccine and screening for all 14 HPV types might result in suboptimal balance of harms and benefits [27].

We found that CIN2 lesions are mainly related to infections caused by more than one genotype. However, our aim focused on the potential role of co-infection in the early phase or progression of lesions and did not find significant differences according to the cytological groups, due to the low rate of severe disease cases. Several studies did not describe a higher risk of severe dysplasia in women with multiple infections compared with women with a single genotype [28,29]. However, further prospective studies could clarify the role of multiple infections identifying the clinically significant impact of specific combinations of HPV genotypes.

The target population could have affected the results on the effectiveness of HPV-vaccination, based on the higher prevalence of infections in the vaccinated group. However, the stratified analysis by HPV-16, the most prevalent and preventable genotype, did not show a significant difference between the vaccinated and non-vaccinated. The type of population enrolled in our study could explain some of the findings’ results. Firstly, the median age of vaccination was high. Real life studies showed that HPV vaccine effectiveness is highest when it is administrated before sexual debut [30]. The American Cancer Society HPV vaccination guidelines (2020) [31], did not endorse the administration in adults aged 27–45 years for limited public health benefit potentially preventing only 0.5% of cancer cases, 0.4% of cervical precancer lesions, and 0.3% of genital warts [32]. The therapeutic role of HPV vaccine is controversial. Our recent study highlighted that vaccine administration could reduce the recurrence rate in women after LEEP [33]. However, a therapeutic role of vaccination was not found in women with a previous HPV-positivity. Prospective studies in naïve women could assess the real-life effectiveness of vaccination in our setting.

The retrospective epidemiological design can raise several concerns. Although the present study reports the main epidemiological findings of the HPV screening program in ASC-US women since the introduction of HPV-DNA in 2016, our data are not representative of the general population, especially for the prevalence and distribution of HPV-genotypes, based on the adherence to the screening program. The small sample size during the follow-up, as well as the heterogeneous follow-up period, did not allow us to clearly identify the prognostic role of several demographic and clinical variables. Moreover, the low numbers of fully vaccinated patients did not allow us to assess the preventive role of the vaccine. Further studies would be needed to assess the epidemiological scenario in the vaccination era.

## 5. Conclusions

Monitoring and genotype identification are crucial to promptly identify cross-protection and type-replacement, particularly after the implementation of a vaccine program. In line with WHO call for action to eliminate cervical cancer as a public health problem [34] future multicenter studies could better estimate HPV risk-factors, as well as those which play a role in the progression of disease to plan more adequate preventive strategies.

## Figures and Tables

**Figure 1 ijerph-19-00693-f001:**
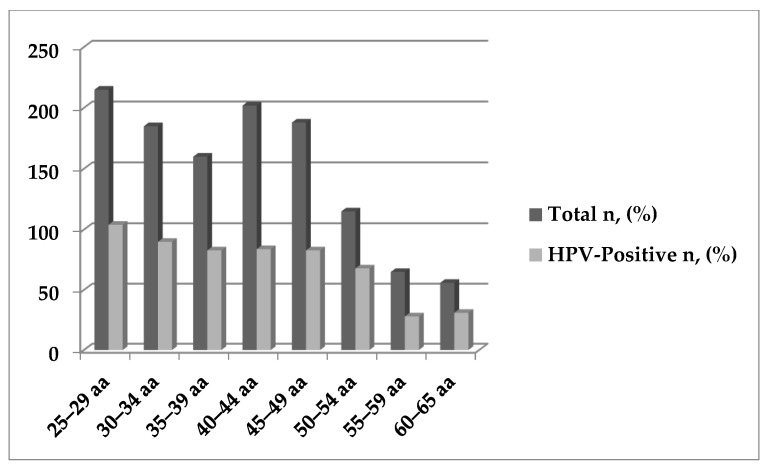
HPV-DNA positivity by age groups.

**Table 1 ijerph-19-00693-t001:** Demographic characteristics of the cohort study.

Median (IQR) age at baseline		41 (31–48)
Civil Status, n (%)	Not declared	366 (30.9)
	Divorced/single	496 (41.8)
	Married	324 (27.3)
Level of Education, n (%)	Elementary school	19 (1.6)
	Middle school	247 (20.8)
	High school	343 (28.9)
	Degree	183 (15.4)
	Not declared	394 (33.2)
Menopause, n (%)		206 (17.4)
Parity, n (%)		633 (53.4)
Full term delivery, n (%)	0	519/1151 (45.1)
	1	239/1151 (20.8)
	2	296/1151 (25.7)
	≥3	97/1151 (8.4)
Abortion, n (%)		214/1180 (18.6)
Use of contraceptive, n (%)		231/394 (58.6)
Number of women underwent to gynaecological surgery, n (%)		105 (8.9)
Gynaecological Intervention during follow-up, n (%)	LEEP	57 (54.3)
	Hysterectomy	4 (3.8)
	Ablative treatment	44 (41.9)
Outcome of intervention, n (%)	CIN1	7 (16.7)
	CIN2	19 (45.2)
	CIN3	14 (33.3)
	Negative	2 (4.8)
Gynaecological intervention pre-T0, n (%)		50/1179 (4.2)
Vaccinated, n (%)		311 (26.2)
Median (IQR) Age at first dose of vaccine		38 (30–47)
Familiarity for breast cancer, n(%)		182 (15.4)
Familiarity for uterus cancer, n(%)		42 (3.5)
Familiarity for ovary cancer, n(%)		8 (0.7)
Familiarity for gynaecological neoplasia, n(%)		4 (0.4)

**Table 2 ijerph-19-00693-t002:** Clinical and epidemiological characteristics at baseline.

Results at Baseline (T0) (n = 1186)		
Presence of at least one genotype hr-HRP, n (%)		571/1186 (48.1)
HPV-16, n (%)		207 (36.3)
HPV-18, n (%)		60 (10.5)
HPV-31, n (%)		107 (18.7)
HPV-33, n (%)		16 (2.8)
HPV-35, n (%)		21 (3.6)
HPV-39, n (%)		59 (10.3)
HPV-45, n (%)		14 (2.5)
HPV-51, n (%)		61 (10.7)
HPV-52, n (%)		55 (9.6)
HPV-56, n (%)		67 (11.7)
HPV-58, n (%)		43 (7.53)
HPV-59, n (%)		38 (6.7)
HPV-66, n (%)		68 (11.9)
HPV-68, n (%)		55 (9.6)
Number of genotypes isolated, n (%)	0	615 (51.9)
	1	393 (33.1)
	2	121 (10.2)
	3	42 (3.5)
	4	13 (1.1)
	5	2 (0.2)
Number of co-infections, n (%)		178 (15.0)
Colposcopy analyses, n (%)	Normal	109/469 (33.2)
	G1	135/469 (41.7)
	G2	52/469 (16.4)
	GSC-NV	25/469 (8.7)
Results of biopsies, n (%)	Low-grade lesion	91/261 (34.5)
	CIN2	35/261 (13.4)
	CIN3	14/261 (5.4)
	CIN0	116/261 (25.245.0)
	Not determined	5/261 (1.9)

**Table 3 ijerph-19-00693-t003:** Logistic regression analysis to assess the relationship between demographic, epidemiological and clinical variables and severity of diseases (CIN2+) at baseline.

Variables		OR (95% IC)	*p*-Value
Age, years		1.01 (0.99–01.03)	0.49
Age groups, years	25–29	Ref	Ref
	30–34	1.67 (0.73–3.82)	0.22
	35–39	0.95 (0.38–2.33)	0.90
	40–44	1.00 (0.43–2.32)	0.99
	45–49	1.00 (0.42–2.38)	0.99
	50–54	1.76 (0.69–4.52)	0.24
	55–60	1.53 (0.51–4.54)	0.45
	>60	0.92 (0.20–4.29)	0.91
Positivity to HPV-DNA at baseline		1.29 (0.64–2.59)	0.48
Aged > 41 years		0.51 (0.31–0.86)	0.01
Education level	Elementary school	-	-
	Middle school	1.09 (0.54–2.21)	0.80
	High school	0.81 (0.44–1.50)	0.50
	Degree	0.39 (0.18–0.87)	0.02
Civil status	Married	0.96 (0.49–1.87)	0.91
Previous Abortion		0.87 (0.43–1.74)	0.69
Pre terms delivery		3.9 (0.4–38.2)	0.24
Parity		0.57 (0.34–0.96)	0.04
Menopause		0.77 (0.35–1.69)	0.51
Use of contraceptive		1.64 (0.58–4.65)	0.35
Gynaecological intervention before baseline		0.72 (0.21–2.53)	0.61
Vaccinated before baseline		1.13 (0.42–3.04)	0.80
Age at first dose vaccination		1.0 (0.97–1.03)	1.00
Co-infection at baseline		1.37 (0.82–2.30)	0.23
Familiarity for cancer	Uterus	1.30 (0.26–6.55)	0.75
	Ovary	-	-
	Breast	0.66 (0.32–1.36)	0.26
	Other gynaecological neoplasia	-	-

## Data Availability

Dataset is available in case it is requested for motivated reasons.

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
