# Peer review of "Cervical Screening in North Sardinia (Italy): Genotype Distribution and Prevalence of HPV among Women with ASC-US Cytology"

_ijerph, 2022, doi:10.3390/ijerph19020693_

Round 1

Reviewer 1 Report

the comments are in the main document that is attached. 

Author Response

We Thank the Reviewer for his/her kindest assessment. Pls find attached the point-by-point response in the attached file

Reviewer 2 Report

To describe the prevalence of HPV infections in North Sardinia we need a population based study including a representative selection of all women in screening age (25-64 years). Most women will have normal cytology. Different cytology diagnosis have different prevalence of HPV and a different distribution of HPV genotypes.

Muresu et al. have included mostly women with cytologic ASC-US. It is best to exclude women with other cytologic diagnoses. Of the 1217 included women, 25 women (2.1%) have ASC-H and 6 women (0.5%) have ACG. If these women are excluded, we have a study population of 1186 women with ASC-US cytology.

There are 330 women who are vaccinated with HPV-vaccine, but only 67 women have taken two doses before inclusion in the study. We do not know the indication of taking av HPV vaccine. When the HPV vaccine cannot prevent a HPV infection the woman already have, it is misleading to present HPV-status in vaccinated and unvaccinated. I think the manuscript should not include any information about HPV vaccinated.

Line 1-4, «Prevalence of HPV infections and epidemiological characteristics of women underwent to cytology examination in North Sardinia» => «Genotype distribution and prevalence of HPV among women in North Sardinia with ASC-US cytology»

Line 20-21, «Aim of the study was to describe the prevalence of HPV infection and epidemiological variables in women underwent to cervical screening in Sardinia, Italy» => «Aim of the study was to describe the genotype distribution, prevalence of HPV infection and epidemiological variables in women in Sardinia, Italy, with ASC-US cytology»

Line 22-23, «Women (25-64 years), with ASCUS or ASCH diagnosis, who underwent a HPV-DNA testing between March 2016-December 2020, were enrolled» => «Included were women (25-64 years), with ASC-US diagnosis, who underwent a HPV-DNA testing between March 2016-December 2020»

Line 30-31, delete «Higher prevalence of infection was found in vaccinated women (73.1% VS. 46.6; p-value<0.001).»

(You cannot present these results. HPV-infections are more common in younger women than older women. There is higher coverage of HPV-vaccine in younger than older women. Are the results age adjusted? More single women chose to take HPV-vaccine than married women. Single women have a higher prevalence of HPV than married women. What was the indication of HPV vaccination? Most of the HPV-vaccinated women in the study were exposed of HPV before vaccination. The vaccine is prophylactic and have no therapeutic effect.)

Line 31-33, delete “Our results support the adoption of a nonavalent HPV-vaccine to prevent infections by rare genotypes and the need of surveillance to implement vaccination programs and preventive strategies”

The nonavalent HPV-vaccine does not prevent rare genotypes. The vaccine covers the seven most common HPV-types in cervical cancer worldwide (HPV 16, 18, 31, 33, 45, 52 and 58). The most detected HPV-types among women with ASC-US was 16, 31, 66, 56 and 51. The nonavalent HPV-vaccine covers only two of the five most prevalent HPV-types in women with ASC-US.

Line 102-104, delete “330 women were vaccinated with a first HPV vaccine dose when their median (IQR) age was 38 (31-47) years. >66% and 27.6% were vaccinated with the tetravalent and nonavalent vaccine, respectively, with 21% who have received two-doses before the screening.”

Delete figure 2 and figure 3. Multiple infections are not interesting. The risk of cancer is dominated by HPV type 16, 18, 45, 33 and 31. Other HPV-types are of less importance.

Line 125-134, delete the paragraph of “Follow-up”. The data is incomplete and outside the scope of this paper.

Line 141-143, delete “Vaccinated women showed a higher prevalence (73.1% VS. 46.6; p-value <0.001), mainly driven by HPV-16 (29.9% VS. 16.8; p-value: 0.006), and HPV-51 (9.0% VS. 3.3%; p-value: 0.01) (Table 6).”

(The data of HPV-vaccinated women are misleading and confusing).

Line 151, “The HPV prevalence was ~48% for women with abnormal cytology” => “The HPV prevalence was ~48% for women with ASC-US cytology”

Line 170-172, delete “It was confirmed the high circulation of some genotypes (i.e., HPV-31, -66, -33, -51) in this Italian region, supporting the recommendation of the nonavalent HPV-vaccine to protect against rare genotypes”

The nonavalent HPV-vaccine covers the seven most common HPV-types in cervical cancer worldwide (HPV 16, 18, 31, 33, 45, 52 and 58), but only one of the four genotypes described in the discussion section.

Line 185-193, delete the paragraph of HPV vaccination. It is misleading and confusing.

Line 196-197, delete “The enrollment of women with a pathological pap smear could have affected our findings, mostly those related to the effectiveness of the HPV-vaccination”

(No. The main problem was that most of the women receiving HPV-vaccination were already infected by HPV. The HPV vaccine is prophylactic, not therapeutic.)

Discussion, add, “In a Swedish study, HPV types 16, 18, 31, 33, 45, or 52 were found in 689 of 808 screendetected invasive cervical cancers (85.3%). Addition of HPV types 35, 39, 51, 56, 58, 59, 66, or 68 (also included in currently used HPV tests) increased prevalence by only 12 of 808 cases (1.5%, for all these 8 types together), (Sundstrom 2020). HPV screening tests in might perform better if restricted to the seven HPV types in the nonavalent vaccine and screening for all 14 HPV types might result in suboptimal balance of harms and benefits (Nygard 2020)”

Line 203-208, delete the conclusion and write something else. The distribution of HPV genotypes in ASC-US cytology is different from the general population.

Table 1, combine parity 3, 4 and 5 into “3 or more”

Table 1, delete “Pre-term delivery” (too small number to be interesting)

Table 1, delete all the lines about HPV-vaccination

Table 2, delete ASCH and AGC. Only women with ASC-US should be included in the manuscript.

Table 2, HR-HRP => hrHPV

Table 2, use CAPITAL LETTERS on HPV (not Hpv).

Table 2, combine “Flogosis” and Negative. Both are normal (CIN0).

Table 2, combine “Condyloma” and CIN1. Both are lowgrade (LSIL).

Delete Table 4. (too small number to be interesting)

Delete Table 5. (all results regarding HPV-vaccination are misleading and confusing)

Delete Table 6. (all results regarding HPV-vaccination are misleading and confusing)

References

Sundström K, Dillner J. How Many Human Papillomavirus Types Do We Need to Screen For? J Infect Dis. 2021 May 20;223(9):1510-1511. doi: 10.1093/infdis/jiaa587. PMID: 32941611.

Nygård M, Hansen BT, Kjaer SK, Hortlund M, Tryggvadóttir L, Munk C, Lagheden C, Sigurdardottir LG, Campbell S, Liaw KL, Dillner J. Human papillomavirus genotype-specific risks for cervical intraepithelial lesions. Hum Vaccin Immunother. 2021 Apr 3;17(4):972-981. doi: 10.1080/21645515.2020.1814097. Epub 2020 Sep 29. PMID: 32990181; PMCID: PMC8018444.

Author Response

(The authors gave the same response as above.)

Round 2

Reviewer 2 Report

All comments have been addressed. The response to the comments is adequate and improved the manuscript.